# A Method of Restraining the Adverse Effects of Grinding Marks on Small Aperture Aspheric Mirrors

**DOI:** 10.3390/mi13091421

**Published:** 2022-08-28

**Authors:** Jiahui Bao, Xiaoqiang Peng, Hao Hu, Tao Lai

**Affiliations:** 1College of Intelligence Science and Technology, National University of Defense Technology, 109 Deya Road, Changsha 410073, China; 2Hunan Key Laboratory of Ultra-Precision Machining Technology, Changsha 410073, China; 3Laboratory of Science and Technology on Integrated Logistics Support, National University of Defense Technology, 109 Deya Road, Changsha 410073, China

**Keywords:** grinding marks, mid-spatial frequency error, elastic adaptive polishing

## Abstract

The grinding method is used as the preliminary processing procedure for small aperture aspheric mirrors. Regular grinding marks produced in the grinding process significantly affect the mid-spatial frequency error; however, because of their small radius of surface curvatures and high steepness, they are difficult to polish using traditional methods. Therefore, in this study, the ultra-precision grinding and polishing process of fused quartz material was investigated, and the influence of grinding marks was analyzed, which achieved the purpose of restraining the grinding marks in the grinding process. The generation mechanisms of horizontal and vertical grinding marks were analyzed by means of simulation and experiment, and the relationship between different grinding process parameters and surface quality was explored. A magnetorheological finishing (MRF) spot method was used to explore the effects of grinding marks on subsurface damage (SSD). The elastic adaptive polishing method was used to polish an aspheric lens with high steepness and small caliber. Based on the principle of an elastic adaptive polishing mathematical model, the grinding marks were suppressed, and the mid-spatial frequency error of the lens was reduced by optimizing the polishing path and composition of the polishing fluid. The final roughness reached 10 nm *Ra*. In this paper, the source of wear marks and their influence on the mid-spatial frequency error of small aperture aspheric mirrors are analyzed, and the grinding marks were suppressed by elastic adaptive polishing.

## 1. Introduction

Small aperture optical aspheric surfaces are widely used in military equipment and optoelectronic communication because of their small aperture and high precision [1,2]. These surfaces are used for different types of systems, such as laser guidance, radar ranging, aviation, aerospace telescopic cameras, infrared thermal imaging engineering, and other optical instruments [3]. They are widely used in micro-aperture aspheric mirrors and are made from all kinds of optical materials [4].

High-precision optical mirrors often need to achieve nanometer accuracy in the full frequency range. Low-frequency errors mainly produce imaging aberration, mid-spatial frequency errors that cause small-angle scattering, flares, and reduce the resolution, making the image blurred; high-frequency errors cause large-angle scattering and energy loss. The mid-spatial frequency errors of optical elements caused by grinding marks have become an urgent problem to be solved in modern optical manufacturing [5,6]. The two most common paths used in existing deterministic polishing techniques are rotation and raster. These periodic polishing tools leave residual errors on the surface of the final optical element [7]. Many optical workers often refer to this residual error as mid-space frequency error or “ripple”. The “ripple” size and periodicity depend on the polishing parameters [8].

Kuriyagawa et al. [9] theoretically analyzed the formation mechanism of nano-topography on axisymmetric grinding surfaces and the relationship between morphology and grinding conditions. The results showed that the vibration of the grinding wheel produced nano-morphology. In addition, the wear marks varied with the speed of the grinding wheel and the speed of the workpiece. Chen et al. [10] analyzed the effects of feed speed, workpiece spindle speed, the synchronous vibration of the grinding wheel, and phase shift on surface quality and optimized the grinding process parameters. The main generation mechanism of the helix around the center of the workpiece was analyzed using simulations and experiments. Moreover, the variable feed speed of the workpiece’s center was modeled, and an approximately straight line of the central area around a circle was revealed. Chen et al. [11] simulated the distribution of grinding points through the trajectory of a single abrasive particle and analyzed the distance relationship between grinding points. Matching the optimized grinding parameters can effectively restrain the generation of grinding lines. Zhang et al. [12] proposed a method for the fast polishing of aspheric surfaces with large-scale polishing tools and summarized the influence of the change in main parameters such as velocity and swing angle on the quality of the aspheric surface. The experimental results showed that the surface roughness and profile quickly converged using large area polishing tools; however, the accuracy obtained using this method depended on the surface profile of the polished aspheric optical elements. Lee et al. [13] used an airbag polishing tool to remove grinding marks and introduced the superposition method of polishing pattern footprints to obtain extended full-contact polishing without considering the aspherical lens profile or tool path control. Yin et al. [14] developed an ultra-precision computer number control (CNC) compound machine tool for small-diameter aspheric optical glass lens molds, which processed aspherical tungsten carbide molds with apertures of 6.6 mm. After oblique axis ultra-precision grinding, the surface roughness reached 6.8 nm roughness (*Ra*) and the shape accuracy reached 383 nm peak and valley (*PV*). After oblique magnetorheological polishing, the surface roughness reached 0.7 nm *Ra* and the shape accuracy reached 221 nm *PV*. Nie et al. [15] developed deformable polishing tools with dents and mixed textures using curing and the pneumatic control of silicone materials. The surface texture of the deformable polishing tool was adjusted according to the stage and surface requirements, and the polishing tool with a smooth or dent texture was made by adjusting the cavity pressure of the double-layer structure. After polishing for 20 min, the surface roughness *Ra* decreased to 10 nm, and the sunken tool showed a higher material removal rate and higher surface roughness reduction rate, which were related to the higher polishing force and pressure of the dented tool. Currently, research is mainly focused on a single process of grinding and polishing. From this focus, the effects of the change in grinding parameters, the formation mechanism of grinding marks, the synchronous vibration of the grinding wheel on the surface shape accuracy, and surface quality were analyzed. The inhibition of grinding marks is carried out using different polishing tools, polishing parameters, and polishing paths in the polishing process. However, systematic research on grinding and polishing is rare, and the aperture of the target optical element is too large or small; therefore, there is a lack of research on high steepness aspheric grinding and polishing processes for small and medium calibers.

Currently, the demand for small aperture optical aspherical mirrors is significant and the precision is high. Small-aperture aspheric glass is molded on an ultra-precision glass molding machine, which is only suitable for glass with amorphous and low melting points, and the processing precision is limited. Traditional grinding, milling, and polishing methods have a long machining cycle, and new surface defects are introduced in the machining process [16,17]. Grinding leaves marks and defects on the machined surface; therefore, subsequent ultra-precision polishing is required to improve the surface quality [18,19]. Owing to the narrow processing space caused by the edge warping of the high-order term of a small-caliber aspheric surface, it was impossible to use wheel magnetorheological, smoothing, and other polishing methods [20,21,22].

In this study, using an established theoretical model and error separation, the generation mechanism and suppression strategy of grinding marks are studied. The effects of grinding parameters on the surface quality and grinding marks on SSD were analyzed using a single-factor experiment and magnetorheological spot method. The elastic adaptive polishing method is proposed to connect the grinding process to further restrain the grinding marks.

## 2. Study on Grinding Marks

### 2.1. Formation Mechanism of Grinding Marks

The grinder used was an ultra-precision grinder developed by the research group. The maximum speed of the grinding wheel spindle was 15,000 rpm, the maximum power was 1200 W, and the rotation accuracy was 1 μm. The spindle of the workpiece was an aerostatic spindle, and its rotation accuracy was 1 μm. Electroplated and resin grinding wheels were used in ultra-precision grinding, and the diameter of the grinding wheel was 40 mm. The material used was ϕ 25 × 30 mm JGS1 fused quartz blank plane parts, and the roughness was approximately 1 μm. The density was 2.2 g/cm^3^, the elastic modulus was 82 GPa, and Poisson’s ratio was 0.17, which met the application conditions. The processed aspheric surface was determined using Equation (1). Its caliber is 25 mm and the vector height is 5.18 mm:(1)$$z=\frac{x^{2}}{R\left[+\sqrt{1-(1+K)x^{2}/R^{2}}\right]}+A_{4}x^{4}+A_{6}x^{6}+A_{8}x^{8},$$
where R = 17.313, K = −1.97, A_4_ = −3.928 × 10^−5^, A_6_ = −5.378 × 10^−8^, A_8_ = 4.671 × 10^−11^.

Its caliber and vector height were measured using a white light interferometer (Zygo NewView 700) under a 5× lens with a scan size of 1.69 × 1.27 mm. In the process of measuring the roughness, as shown in Figure 1, it was found that there were two kinds of grinding marks: horizontal grinding marks, which were distributed horizontally, and vertical grinding marks, which were distributed vertically.

The workpiece and the grinding wheel spindles of the machine tool used in this experiment were arranged perpendicular to each other, and the friction between each diamond particle on the grinding wheel and the workpiece surface was removed accordingly. Owing to the uneven size of diamond particles and the inconsistent size of diamond particles exposed under the substrate, the scratch depth of each diamond particle was different, which was completely copied on the surface of the workpiece, resulting in grinding marks. The matching relationship between the workpiece and grinding wheel speeds was the main cause of grinding marks. To analyze the mechanism of grinding marks and the influence of grinding marks on the surface shape and roughness, we simulated the grinding marks produced by a single abrasive material in the process. Given the coordinates of grinding points ($x_{i}$,$y_{i}$,$z_{i}$) [11], the distribution expression of the grinding points of a single diamond abrasive is as follows:(2)$$\{\begin{array}{l} x_{i}=r_{i}\cdot \cos\left(\omega_{w}\cdot t_{i}\right)\\ y_{i}=r_{i}\cdot \sin\left(\omega_{w}\cdot t_{i}\right)\\ z_{i}=f\left(r_{i}\right) \end{array},$$
where $f\left(r_{i}\right)$ is an aspheric function, and the wear particles produced grinding mark points in a cycle. The single period is as follows:(3)t=1 / N,
where N is the speed of the grinding wheel. The radius of the small diameter aspheric surface is L. F is the feed speed in the X-direction. The single processing cycle is as follows:(4)T=L / F,
where $n_{t}$ is the total number of grinding mark points in a single cycle of wear particles:(5)$$n_{t}=T\;/\;t,$$
where $t_{i}$ is the time required for the generation of a single grinding mark, and *i* is the step length of the number of points in a single cycle, which ranged from $0\sim n_{t}$:(6)$$t_{i}=\frac{60}{n_{t}}\cdot i.$$$r_{i}$ is the radius at the grinding point (mm), as is found as follows:

(7)$$r_{i}=L-f\cdot t_{i}.$$$n_{w}$ is the rotational speed of the workpiece, and *w_w_* is the angular speed of the workpiece; their relationship is found as follows:



(8)
$$\omega_{w}=\frac{n_{w}}{60}\cdot 2\pi .$$



Nine groups of grinding parameters are listed in Table 1, and the simulation results of the parameters and the surface morphology of the high-resolution optical microscope and interferometer are shown in Figure 2.

From the simulation results, the surface morphology was measured using a high-resolution optical microscope (KEYENCE VHX3000) under a 50× lens and the surface shape of an interferometer (Six-Inch Aspheric Vertical Interferometer of Zygo VerFire). The simulation of the grinding marks of single abrasive particles was consistent with that of the actual machining results, and the horizontal grinding marks produced by abrasive particles formed spiral grinding marks. The distributions of abrasive grinding marks were mainly related to the speed of the grinding wheel and workpiece, and the densities of the abrasive grinding marks were mainly related to the feed speed. This distribution was not affected by other factors, such as grinding depth and workpiece material.

### 2.2. Mechanism of Periodic Ring Grinding Marks

The vertical grinding marks significantly affected the mid-spatial frequency error of the workpiece. To analyze the source of the vertical grinding marks, we measured the straightness of the *X*-axis guide. As shown in Figure 3a, the sensor used in the measurement was STIL’s CL2-MG140 laser spectral confocal displacement sensor, which had a resolution of 10 nm and an accuracy of 80 nm. Before measuring the straightness, the performance of the sensor was calibrated. Under a static condition of the confocal sensor, the sensor recorded a numerical jump of approximately 50 nm, which showed that, under the static condition, there was no noise in the process of data acquisition and processing, and the straightness measurement condition was available. The flat crystal used for the measurement was a plane machined using an ion beam. As shown in Figure 3b, its *PV* was 8.5 nm and the root mean square (*RMS*) was 0.25 nm, which met the measurement requirements and was approximated as an absolute plane.

The measurement results are shown in Figure 4. To verify the accuracy of the measurement process, we measured the straightness of the machine tool in the *X* direction. To distinguish the accuracy of the two measurements, we moved the machine tool in the *Z* direction by 0.3 μm before the second measurement. The two measurement results were approximately the same. Using the comparative analysis of the separation error, the machine tool had a periodic vibration in the *X* direction, which was the root cause of vertical grinding marks and led to the grinding marks of lenses in periodic rings.

### 2.3. Central Crushing Caused by Abrasive Grinding Marks

In addition to the two grinding marks on the edge, there was a broken ring in the central part of the workpiece. The image in Figure 5a was measured using a high-resolution optical microscope (KEYENCE VHX3000) under a 50× lens, and the image in Figure 5b was measured using a white light interferometer (Zygo NewView 700) under a 5× lens with a scan size of 1.69 × 1.27 mm, which showed that periodic ring lines suddenly occurred near the center of the workpiece, and the grinding marks were deep and broken near some of the rings.

To analyze the causes of crushing in the central ring of the workpiece, a single grinding particle in the center of the workpiece was simulated, and its parameters were from experiments 1-4, as listed in Table 1 and shown in Figure 6.

With the grinding wheel gradually feeding to the center, the distribution of wear particles changed abruptly, and the radial distance of wear particles was close to the axial distance, showing a circular distribution. The feed speed of the grinding wheel was constant. When the speeds of the grinding wheel were 30 or 40 rpm, the wear distribution of particles was more concentrated, and multiple abrasive trajectories overlapped in the center of the workpiece, resulting in the repeated machining of the center, overlapping grinding marks of multiple abrasive grains, and squeezing each other. This process resulted in the continuous widening of grinding marks near the center of the workpiece and increasing depth. When the workpiece speed was 50 rpm, the grinding marks showed a periodic spiral distribution, the wear particle trajectory distribution was more uniform, the overlapping part was small, the broken part was reduced, and the material was removed evenly.

The measurements were performed using a white light interferometer under a 20× lens with a scan size of 0.47 × 0.35 mm. One sampling point was taken every 5 mm, and the average values of the two groups were measured in the *X* and *Y* directions. The specific results are shown in Figure 7. Moreover, the roughness decreased gradually from the center to the edge of the workpiece. The reason for this result was that near the center of the workpiece, the central part was repeatedly crushed, owing to the overlap of the marks of the wear particles. Moreover, the roughness of the middle part was poor, whereas, in the part far away from the center, the grinding marks distribution was more uniform. This was because the measured area was small and the middle- and low-frequency error changes were not included in the roughness.

### 2.4. Grinding Marks Suppression Strategy

The simulation results showed that the grinding lines of different workpiece speeds were different, and different rotational speeds formed different regular patterns. When the rotational speed of the workpiece was 60 rpm, the gap between the grinding points at the same distance from the center of the circle was larger, and the residual height between the points was larger. The results of the high-resolution optical microscope showed that the grinding marks were more obvious (mostly plowing scratches), and the surface was broken more. When the rotational speed of the workpiece was 50 rpm, the interval between the grinding points within the same distance from the center of the circle was small. Moreover, the distribution of grinding points on the surface shape was more uniform. The results of the high-resolution optical microscope showed that plowing scratches were less, crushing was shallow, the surface morphology was better, and material removal was more uniform. When the feed speed was reduced, the number of wear particles in the simulation diagram increased; however, it did not change the periodic distribution pattern of grinding marks. To verify the influence of grinding parameters on the surface shape accuracy, we conducted the experiment according to the experimental design listed in Table 1. The speed of the grinding wheel was increased and the results are shown in Figure 8a. The grinding marks changed; however, the surface shape did not improve with an increase in the speed of the grinding wheel. As shown in Figure 8b, neither reducing the feed speed nor increasing the feed speed improved the surface shape accuracy. According to the first two groups of experiments, the best feed and grinding wheel speeds were selected to change the workpiece speed. From the five groups of experimental results shown in Figure 8c, matching the workpiece speed and other grinding parameters was an important factor affecting the surface shape accuracy. When the rotational speed of the workpiece was 50 rpm, the *PV* and *RMS* values were the smallest. The simulation results of single wear particles showed that the distributions of wear particles were the most uniform, material times were removed uniformly, and there were no significant fluctuations between points; therefore, the *PV* values were smaller. The results of the high-resolution optical microscope showed that the surface morphology of the workpiece was acceptable, there was no significant fragmentation, plowing scratches were shallow, and the changes in surface shapes were small. The binary treatment of the crack showed that when the rotational speed was 50 rpm, the density of the broken surface was the smallest.

From the aforementioned experimental results, speed matching was the root cause of periodic spiral grinding marks distribution in the polar shaft machining process, and increasing the grinding wheel speed and reducing the feed speed did not improve the surface shape accuracy. Only when the grinding parameters reached the best matching parameters, the distribution of abrasive grinding marks was uniform, surface crack densities were small, and surface shape accuracies were the highest.

The *X* direction was the sensitive direction of the ultra-precision grinder, which had the largest stroke in the machining process, which was very important for the surface shape accuracy. From the results in Figure 4, the machine tool had periodic vibration in the *X*-direction, resulting in periodic ring grinding marks. This significantly affected the medium and high-frequency errors of the lens, in which the straightness error of the *X*-axis in the *Z* direction was up to 0.6 μm/50 mm. At this time, the machining affected the surface shape accuracy, and medium and high-frequency errors were introduced. To analyze the influence of this type of vibration, we simulated and analyzed the surface shape accuracy before and after compensation. As shown in Figure 9a, the *PV* and *RMS* values converged after compensation.

To verify the simulation results, the actual machining experiments after the compensation section were conducted, as shown in Figure 9b. Because the grinding wheel was approximated as an ellipsoidal contact rather than a point contact during the machining process, the actual machining results were not the same as those of the simulation results; however, the power spectral density (PSD) curve analysis of the surface shape before and after machining was conducted. As shown in Figure 10, it was found that the continuity of the shape after high precision compensation was acceptable. Moreover, the medium and high-frequency errors were significantly reduced, and after compensation, the *PV* value was reduced, and the subsequent polishing margin was reduced.

## 3. Evolution of Surface and Subsurface Characteristics in the Grinding Process

### 3.1. Influence of Grinding Parameters on Surface Quality

The grinding conditions used are listed in Table 2. In this experiment, the effects of grinding lines caused by the type of grinding wheel, cutting depth (*A*_e_), and cutting speed (*V*_c_) on surface shape accuracy and surface quality were studied.

Figure 11 shows the surface roughness morphology of fused quartz at different cutting depths for a 1000# electroplated grinding wheel, which contained three kinds of surface quality information (*PV*, *RMS*, and *Ra*). When the cutting depth was 1 μm, *PV*, *RMS*, and *Ra* reached 3362, 173, and 122 nm, and the surface was mainly composed of shallow periodic grinding marks. With an increase in cutting depth, the surface quality deteriorated gradually. When the cutting depth was 10 μm, *PV*, *RMS*, and *Ra* reached 4197, 216, and 156 nm. For this scenario, under a white light interferometer, we saw that the surface was broken, pits, plowing scratches, some pixels were lost, and the surface quality was poor. From the comparison in Figure 12, with an increase in cutting depth, the grinding marks became more obvious, and the surface quality increased with an increase in cutting depth.

Figure 13 shows the surface morphology of fused quartz under a white light interferometer at different wheel speeds. The changes in surface roughness *PV*, *Ra*, and *RMS* of fused quartz at different wheel speeds are shown in Figure 14. The surface topography was different at different cutting speeds. When the cutting speed was 17.8 m/s, surface quality *PV*, *RMS*, and *Ra* reached 3900, 110, and 83 nm, respectively, and the peak and valley values of deep wear marks were larger. With an increase in cutting speed, the grinding marks became shallower and the peak and valley values decreased in Figure 13b–d. With the continuous increase in cutting speed, the indices of roughness decrease significantly. At a cutting speed of 30.4 m/s, the surface quality of *PV*, *RMS*, and *Ra* decreased to 1978, 83, and 55 nm, respectively. An increase in cutting speed contributes to a smoother surface. When the cutting speed reaches 30.4 m/s, the surface microcracks disappear and the smooth surface produces shallow grinding stripes, indicating that the ductile cutting of abrasive particles plays a dominant role in the surface morphology.

Figure 15 shows the surface quality of fused quartz at different feed speeds, and the surface quality was different at different feed speeds. When the feed speed was 1.5 mm/min, *PV*, *RMS,* and *Ra* reached 1969, 101, and 73 nm, respectively, and the roughness was the lowest, periodic grinding marks were close to each other, there were two grinding marks, and the width of a single grinding mark was narrow. With the increase in feed speed, when the feed speed was 3 mm/min, *PV*, *RMS,* and *Ra* were 5636, 125, and 97 nm, respectively, and the roughness was the highest, the measuring interface only showed single grinding marks, the grinding marks were wide, and the peak and valley values were significant. From this, with the continuous increase in feed speed, the surface quality continued to deteriorate. However, from Figure 16, with a decrease in feed speed, the improvement of the surface quality slows down, and reducing the feed speed to improve the surface quality reduces the processing efficiency, which is not desirable.

### 3.2. Effect of Grinding Marks on SSD

In ultra-precision grinding, the maximum depth of the subsurface defects of fused quartz material is the median crack caused by brittle fracture. In the process of abrasive scratching removal, the lateral crack caused by the median crack extending to both sides extends to the surface of the specimen, which will lead to the brittleness removal of the material and form the surface roughness on the surface of the specimen. The crack in the grinding process does not necessarily extend to the surface, and the median crack is not visible on the surface. On the other hand, the median crack constitutes subsurface damage. The existence of these damages not only affects the imaging quality, coating quality, and service life of quartz glass but also directly determines the removal amount and processing efficiency of the next polishing process. The rapid and accurate detection of subsurface damage introduced in the grinding process can guide the optimization of processing technology and improve the processing quality and efficiency of quartz glass.

The size of the diamond grains exposed on the outer surface of the grinding wheel was different, which led to the peak and valley value of the grinding marks, and the different surface quality at the peak and valley, which inevitably led to the change in the depth of the SSD layer. To explore the effect of grinding marks on SSD, an MRF spot method was used to study SSD. As shown in Figure 17, each sample surface was polished using an MRF to form two wedges containing all SSD. The sample surface was etched with 5% hydrofluoric acid (HF) for 10–15 min, the Beilby layer was removed, and the cracks were distinguished at different depths along the surface wedge at 500× magnification using a high-resolution optical microscope (KEYENCE VHX3000).

The grinding parameters are shown in the fifth group of experiments listed in Table 1, when the surface quality parameters *PV*, *RMS*, and *Ra* reached 2911, 137, and 82 nm, respectively. Using a high-resolution optical microscope with a 500× lens to observe the edge of the magnetorheological spot, the SSD along the grinding marks are deep into the lens, the grinding marks lead to a deeper damage layer, and this type of layer damage will significantly affect the subsequent polishing level.

A self-developed high-precision three-dimensional profiler (shown in Figure 17b) was used to measure the midline profile of MRF spots. Figure 18 shows the extension of grinding marks under MRF spots, which are deeply embedded in the subsurface. The position coordinates of the contours are shown in Figure 19a, and the position information corresponding to different depths was obtained. According to the magnetorheological spot profile, the subsurface depth corresponding to different distances of the platform movement and the distance of the platform movement when the crack had just disappeared—that is, the maximum subsurface crack depth—were obtained. On this basis, the images of subsurface cracks with different depths were processed using a binary threshold; the crack in the image area was white and the non-crack matrix around the crack was black. The contrast at this depth was obtained using the ratio of crack to image areas, which quantitatively characterized the subsurface crack density of different depths and finally obtained the evolution law of subsurface crack density along the depth direction. As shown in Figure 19b, the subsurface crack density is distributed exponentially with depth.

As shown in Figure 20, along the direction of the magnetorheological spot, where the crack was located from 0 to 2.8 μm, the subsurface crack density at the peak of the grinding marks was smaller, whereas the subsurface crack density of the grinding marks at the bottom of the valley was higher, and the cracks were continuous. When it was located at 3.3 μm, the subsurface crack at the peak disappeared, whereas the subsurface crack at the valley bottom still existed, and the subsurface crack did not disappear until it was located at the subsurface at 6.2 μm. The existence of grinding marks improves the SSD layer and significantly affects the polishing accuracy and efficiency.

## 4. Removal of Grinding Marks Using Elastic Adaptive Polishing

### 4.1. Mathematical Model of Elastic Adaptive Polishing 

In this study, an elastic polishing tool was proposed, even with a high-density damping cloth and high elastic sponge combined with cerium oxide liquid, as shown in Figure 21. The high elastic sponge provided acceptable support, cerium oxide liquid improved the removal efficiency, and the high-density damping cloth made the polishing tool better fit the grinding mark surface. This method effectively removed the residual grinding marks on the grinding surface and significantly shortened the period of pre-polishing. The size of the polishing area was controlled by the depth at which the elastomer was pressed onto the workpiece, whereas the polishing direction and speed were controlled independently by the tool and rotation speed. Compared with traditional polishing, the elastic adaptive polishing process based on trajectory was an adaptive polishing process with some advantages. First, the elasticity of the main body of the polishing tool allowed it to conform to the grinding marks of the aspheric surface, which could not be realized using traditional polishing processes. Second, there was no hard contact between the polishing tool and workpiece, which minimized the surface divergence. Third, by adjusting the feed speed (equivalent to changing the residence time), the grinding marks on the workpiece surface were uniformly removed on the complex surface; thus, a high-quality surface was obtained. In addition, the process did not require particularly strict machine stiffness and profile performance. This was because the elastomer could “absorb” the irregularity of the tool path or workpiece, form messy scratches in the contact area, and remove grinding marks.

The material removal process used the Preston equation, and the polishing tool kept acceptable contact with the machined optical mirror under the action of bias. In the process of polishing according to the polishing track, the polishing tool moved on the mirror at a relative speed *V*, at any time *t*, the contact area between the polishing tool and the mirror was *Zc*, and the corresponding pressure distribution was *P_c_*. Subsequently, the material removal efficiency *E_M_* of the elastic grinding wheel was calculated using the following:(9)$$E_{M}(x,y,t)=k_{preston}\cdot P_{c}(x,y,t)\cdot V(x,y,t).$$

After the time *t*, the material removal of the whole mirror was as follows:(10)$$M(x,y)=\int_{t}E_{M}(x,y,t)dt.$$

When the polishing tool was elastically deformed to adapt to the grinding marks on the optical mirror, according to the elastic theory, the pressure *P_h_* at the high point of the mid-spatial frequency error was greater than the pressure *P_l_* at the low point. The ratio of material removal efficiency at the high and low points of grinding marks was calculated as follows:(11)$$r_{m}=\frac{E_{M}{}_{h}}{E_{M}{}_{l}}=\frac{k_{Preston}\cdot p_{h}\cdot v_{h}}{k_{Preston}\cdot p_{l}\cdot v_{l}}.$$
where *E**_Mh_* and *E**_Ml_* were the material removal efficiencies of the high and low points of the grinding marks, respectively; *k_Preston_* was the Preston constant; and *v**_h_* and *v_l_* were the relative polishing speeds of the high and low points of the optical mirror, respectively. Under the same process conditions, it was considered that the *k_Preston_* was the same everywhere, and the velocity gradient of the high and low points of the grinding marks approached 0; therefore, it was simplified as follows:(12)$$r_{m}=\frac{p_{h}}{p_{l}}.$$

The ratio of material removal efficiency at the high and low points of the optical mirror was equal to the ratio of pressures. If the peak-valley value of mid-spatial frequency error (*PV*) before polishing was defined as *pv_MSFR_* and the peak-valley value of mid-spatial frequency error after fairing was *pv_MSFR_*′, the following logic held true in the polishing process studied in this work:(13)$$r_{m}>1\Leftrightarrow pv_{MSFR}>pv_{MSFR}{}^{\prime }.\;$$

The material removal efficiency of the optical mirror’s high point was higher than that of the low point, which was a necessary and sufficient condition to suppress the grinding marks; therefore, it was also called the basic condition of restraining the grinding marks. Furthermore, the convergence rate *s_c_* and convergence ratio *r_c_* of the mid-spatial frequency error peak and valley (*PV*) were calculated as follows:(14)$$s_{c}=k_{Preston}\cdot \Delta p\cdot v,$$
(15)$$r_{c}=\frac{pv_{MSFR}}{pv_{MSFR}{}^{\prime }}.$$

The pressure difference between high and low points was calculated using the following: $\;\Delta p=p_{h}-p_{l}$.

### 4.2. Polishing Path Optimization Strategy

For some wear particles, the time sequence of engagement with the workpiece went through the following stages: proximity, friction, plowing, and cutting, as shown in Figure 22. If the pressure acting on the removal trajectory of wear particles is different, the effect is also different. If the pressure is small, the trajectory occurs only through friction and plowing; therefore, the wear particles will not have cutting behavior. To achieve fast conformal polishing, the contact force between the polishing tool and the workpiece is relatively large and uniform. Subsequently, the relationship between the forward pressure and the workpiece trajectory in the extended macro-scale tool-workpiece contact model was analyzed, and the best machining trajectory was distinguished according to the results by finally calculating the positive pressure of each position.

In the process of rapid polishing, the polishing tool and the workpiece rotated, and while the workpiece rotated, it also moved in the *X* and *Z* directions, according to the change in the curvature of the aspheric surface. The change in the *Z* direction mainly changes with the change of X; therefore, the movement distance of X is the most important factor affecting the forward pressure. As shown in Figure 23 and Figure 24, when the displacement was 12 mm, the pressure distribution from the center to the edge of the workpiece varied significantly. This is because when the workpiece moves to the edge position of the polishing tool, there is a large difference in the height between the polishing workpiece and the part in contact between the center and the edge of the workpiece, which leads to edge warping and affects the polishing effect at the edge. When the displacement is 8 mm, the pressure distribution on 0–8 mm is more uniform, and the polishing effect is ideal.

### 4.3. Performance Analysis and Optimization of Polishing Fluid

Owing to the significant error of the initial surface roughness after grinding, a 50% proportion of cerium oxide liquid was used for rough polishing. This is because the removal rate increases rapidly with the increase in abrasive concentration and depth. Polishing was conducted according to the best tool path obtained from the 4.2 pressure measuring part, in which the surface feed of the polishing tool was adjusted according to the shape error of the parts (that is, the feed speed of the high steepness lens was faster, and the feed speed of the low degree lens was slower). The process parameters are listed in Table 3.

As shown in Figure 25b, after rough polishing several times, the roughness reached 20 nm, and it was difficult to continue polishing to significantly improve the surface properties. Moreover, the high concentration of polishing liquid had significant damage to the surface shape; therefore it was difficult to realize conformal processing. Therefore, after rough polishing, the concentration of the polishing solution changed to 33%, and the depth of the polishing tool was reduced to 5 mm. At this time, the result, as shown in Figure 25c, reached 10 nm and meets the user requirements.

Ultra-precision grinding has played an important role in the processing of the optical elements of hard and brittle materials. The high-precision surface shape can be obtained by grinding, which can greatly reduce the machining allowance for subsequent polishing, so as to improve the overall processing efficiency of optical parts. However, grinding marks will be produced after ultra-precision grinding, and the surface quality *PV*, *RMS*, and *Ra* can only reach 2,212and 114 nm. The use of a fine-grained grinding wheel will affect the surface shape accuracy because of the rapid wear of the grinding wheel. Therefore, it is necessary to use elastic adaptive polishing to improve surface quality. After path optimization, liquid optimization, and other measures, the surface quality *PV*, *RMS*, and *Ra* reached 1203, 16, and 10 nm, which greatly improved the surface quality. The imaging quality and coating quality were optimized.

In the process of rough polishing, the concentration of polishing particles is high, and the polishing force is large; therefore, the removal effect of polishing particles is significant, mainly in the semi-plastic zone, and some short-band errors are mainly suppressed. In fine polishing, the concentration of polishing particles is low, and the polishing force is small, which is mainly plastic removal, which can restrain the long-band error. As shown in Figure 26, through rough polishing and fine polishing, the full-band convergence of mid-spatial frequency error is realized, and the *Ra* meets the requirements.

## 5. Conclusions

In this study, a small-diameter aspheric surface was polished using ultra-precision grinding and elastic adaptive polishing, and the following conclusions were drawn:(1)Abrasive grinding marks and guideway errors were the root causes of grinding marks, periodic ring grinding marks were suppressed by compensation, and abrasive grinding marks were mainly affected by the matching of the workpiece and grinding wheel speeds. When the best matching ratio was reached, the distribution of grinding marks was uniform, and the surface shape accuracy and surface quality were the best. The overlapping grinding marks of abrasive particles and repeated rolling produced ring breakage at the center of the workpiece.(2)Increasing the grinding wheel speed, reducing the feed speed, and reducing the grinding depth improved the surface quality of the grinding process. According to the experimental conclusions and requirements of different grinding parameters, the experimental scheme was designed to improve the grinding efficiency and improve surface quality. The subsurface cracks extended to a deeper depth along the bottom of the grinding mark valley, and when it reached a certain depth of the subsurface, there was only a subsurface crack in the corresponding position of the grinding marks, which had a significant impact on the subsequent polishing and use of the workpiece.(3)In this study, an elastic adaptive fast polishing method was proposed, which suppressed the grinding marks produced in the grinding process, realized the fast polishing of the mirror surface, and solved the problem of the fast polishing of high-steepness and small-diameter aspheric surfaces. Moreover, the polishing process did not introduce new mid-spatial frequency errors.

## Figures and Tables

**Figure 1 micromachines-13-01421-f001:**
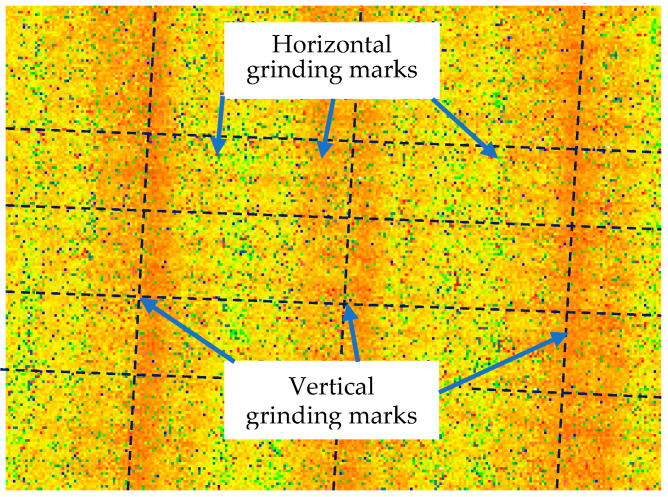
Two different types of grinding marks.

**Figure 2 micromachines-13-01421-f002:**
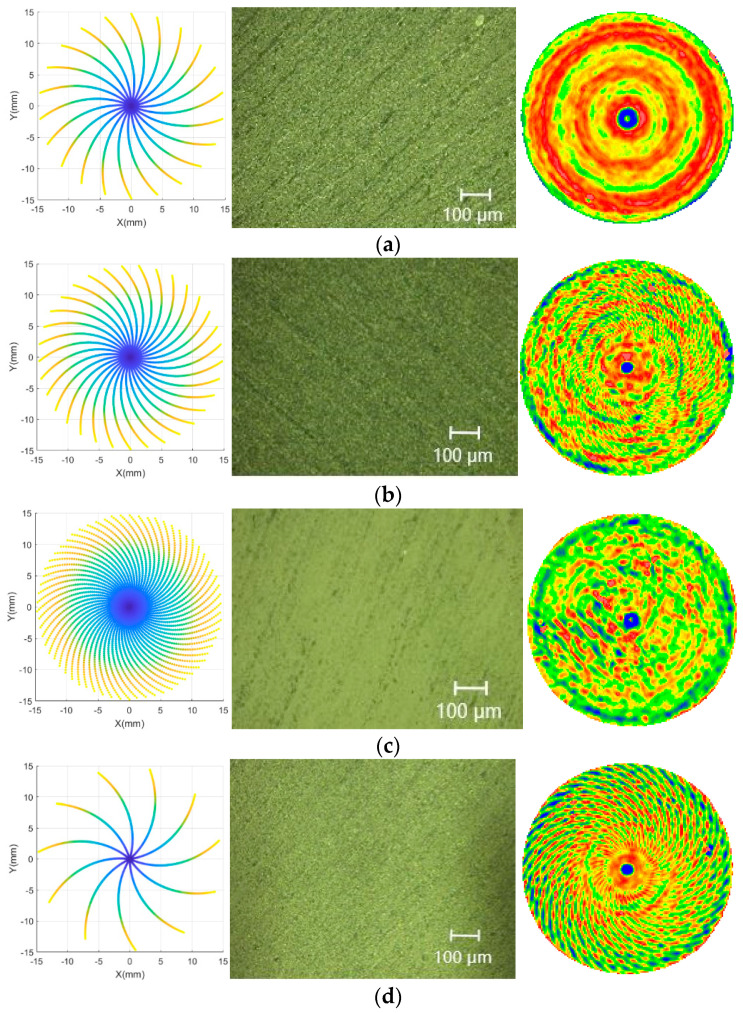
Matching results of different parameters. (**a**) Grinding wheel speed of 2635 rpm, workpiece speed of 30 rpm, and feed speed of 3 mm/min. (**b**) Grinding wheel speed of 2635 rpm, workpiece speed of 40 rpm, and feed speed of 3 mm/min. (**c**) Grinding wheel speed of 2635 rpm, workpiece speed of 50 rpm, and feed speed of 3 mm/min. (**d**) Grinding wheel speed of 2635 rpm, workpiece speed of 60 rpm, and feed speed of 3 mm/min. (**e**) Grinding wheel speed of 2635 rpm, workpiece speed of 70 rpm, and feed speed of 3 mm/min. (**f**) Grinding wheel speed of 2721 rpm, workpiece speed of 50 rpm, and feed speed of 3 mm/min. (**g**) Grinding wheel speed of 2871 rpm, workpiece speed of 50 rpm, and feed speed of 3 mm/min. (**h**) Grinding wheel speed of 2635 rpm, workpiece speed of 50 rpm, and feed speed of 2 mm/min. (**i**) Grinding wheel speed of 2635 rpm, workpiece speed of 50 rpm, and feed speed of 4 mm/min.

**Figure 3 micromachines-13-01421-f003:**
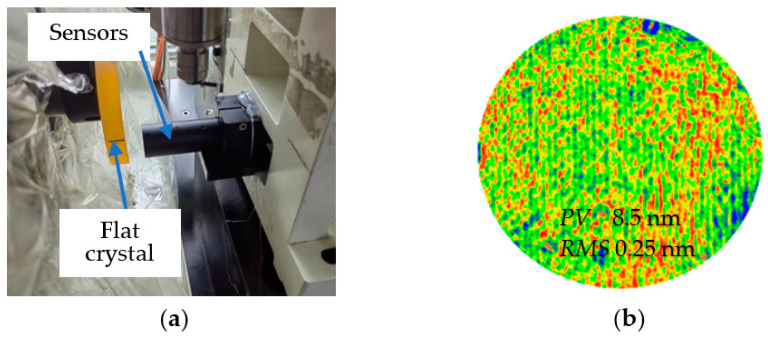
Periodic vibration measurement. (**a**) Mode of measurement. (**b**) Accuracy of flat crystal shape.

**Figure 4 micromachines-13-01421-f004:**
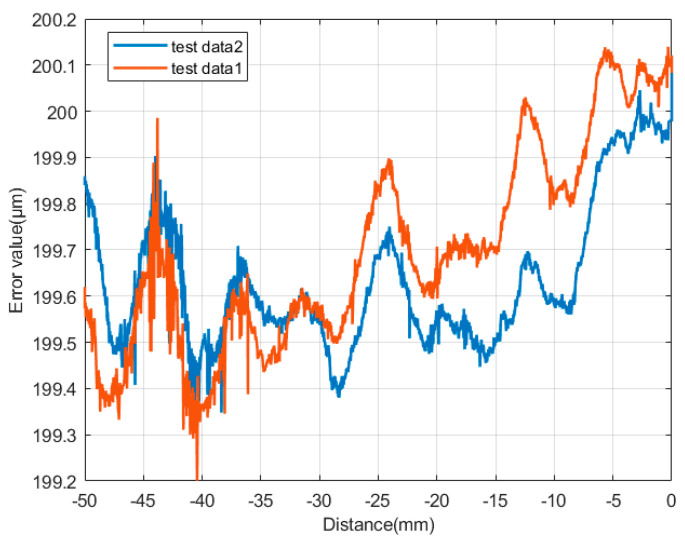
Straightness of the *X*-axis guide.

**Figure 5 micromachines-13-01421-f005:**
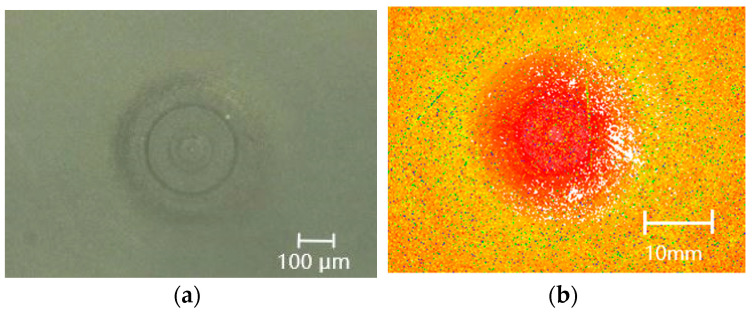
Periodic ring breaking. (**a**) Results of the super depth-of-field microscope. (**b**) Results of the white light interferometer.

**Figure 6 micromachines-13-01421-f006:**
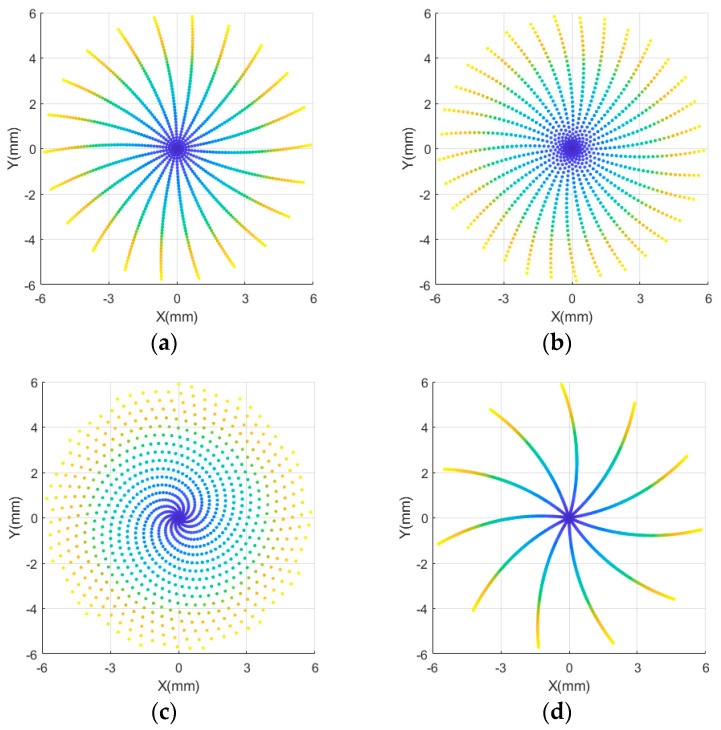
Tracking of abrasive particles in the grinding center under different workpiece matching speeds. (**a**) r30, (**b**) r40, (**c**) r50, (**d**) r60.

**Figure 7 micromachines-13-01421-f007:**
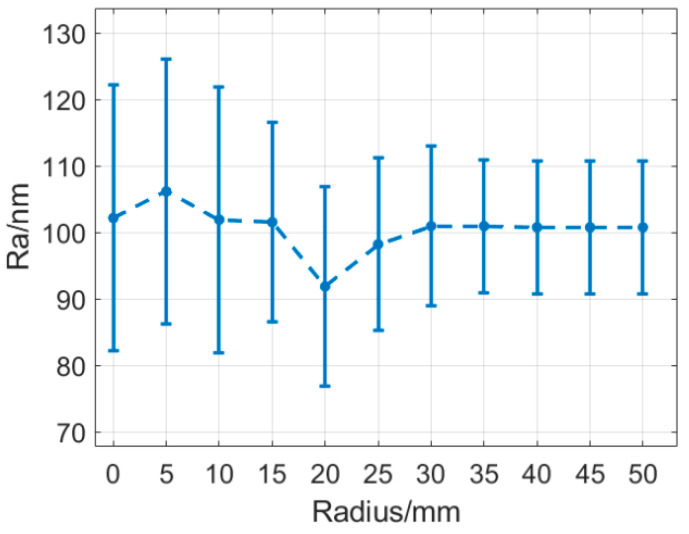
Change in roughness from the center to the edge of the workpiece.

**Figure 8 micromachines-13-01421-f008:**
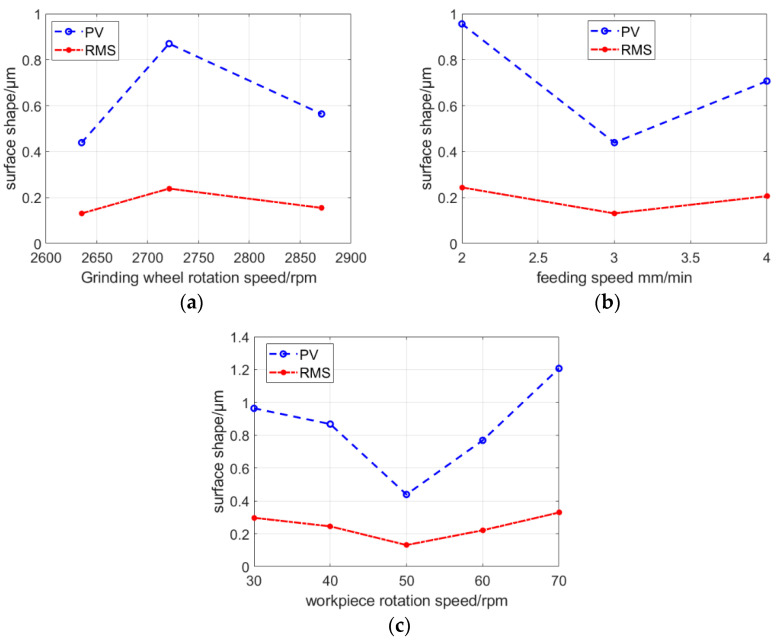
Influence of different parameters on face shape. (**a**) Relationship between surface shape accuracy and grinding wheel speed. (**b**) Relationship between surface shape accuracy and feed speed. (**c**) Relationship between surface shape accuracy and workpiece speed.

**Figure 9 micromachines-13-01421-f009:**
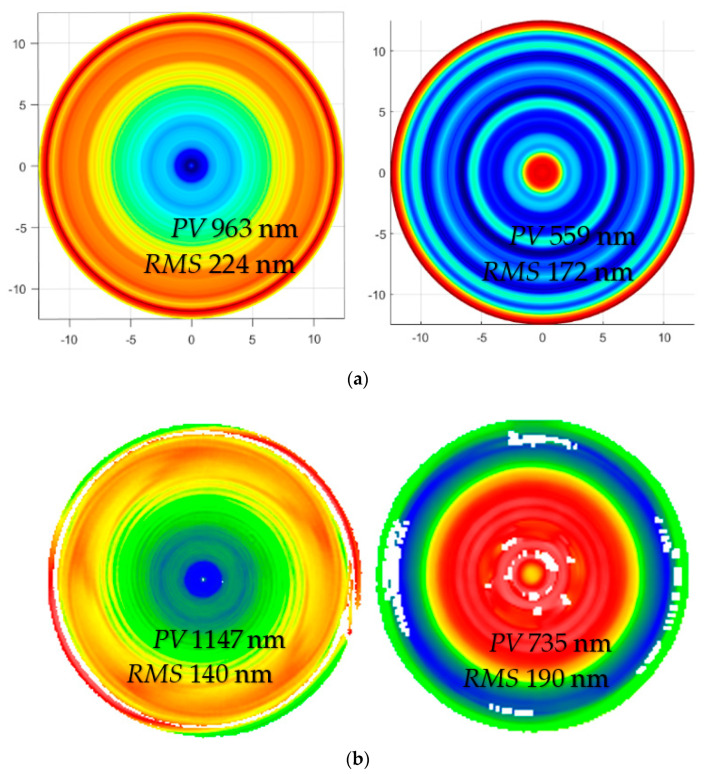
Surface shape simulation and actual machining results. (**a**) Simulation results. (**b**) Actual processing results.

**Figure 10 micromachines-13-01421-f010:**
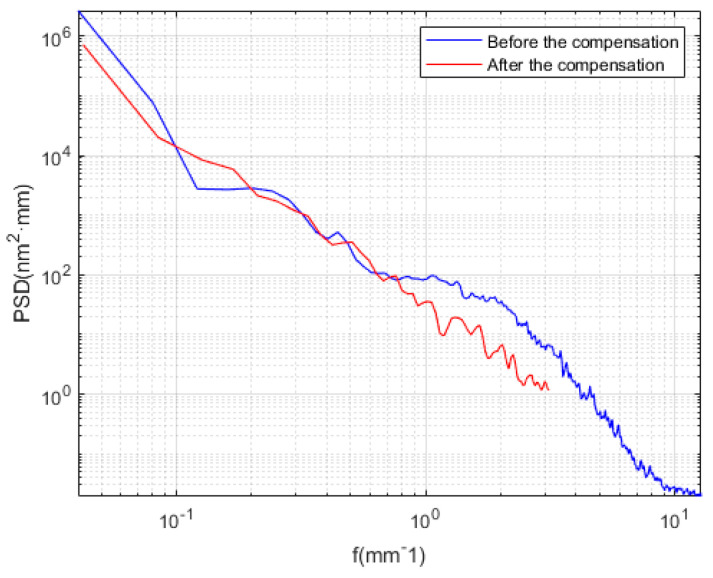
Comparison of particle size distribution curves before and after compensation.

**Figure 11 micromachines-13-01421-f011:**
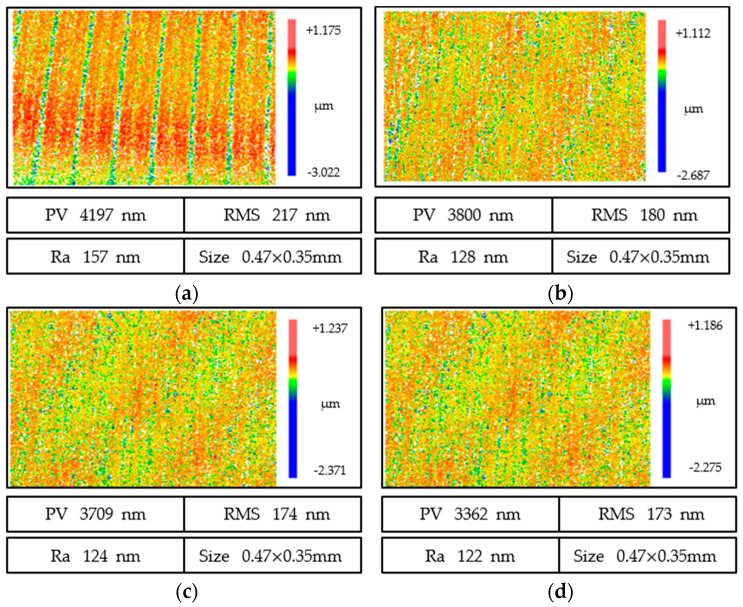
Evolution of the surface quality of grinding marks under different cutting depths for an electroplated grinding wheel. (**a**) *A*e = 10 μm. (**b**) *A*e = 5 μm. (**c**) *A*e = 3 μm. (**d**) *A*e = 1 μm.

**Figure 12 micromachines-13-01421-f012:**
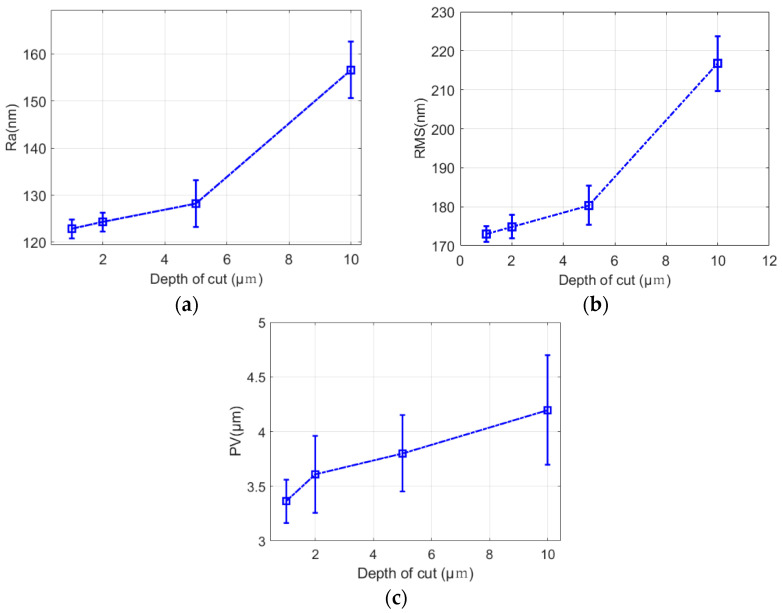
Effects of grinding marks on surface quality for different cutting depths. (**a**) *Ra*. (**b**) *RMS*. (**c**) *PV*.

**Figure 13 micromachines-13-01421-f013:**
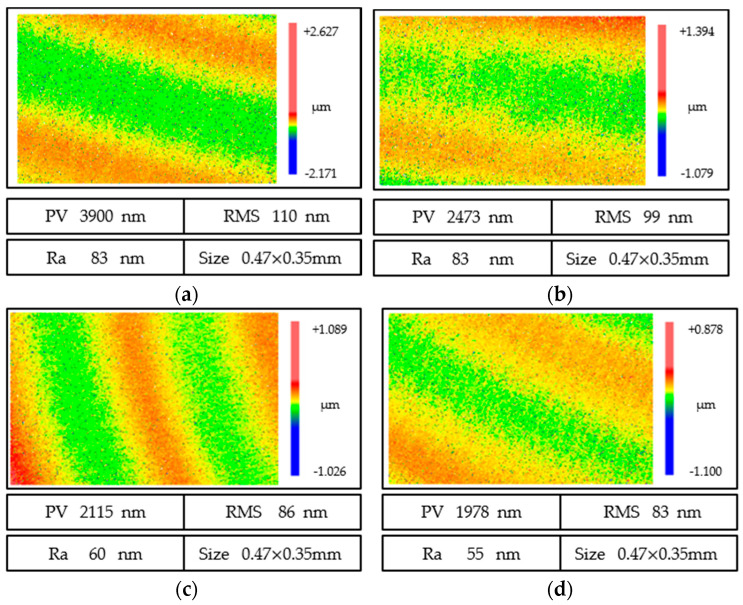
Surface quality evolution of ground fused quartz at different cutting speeds. (**a**) *V*_c_ = 17.8 m/s. (**b**) *V*_c_ = 21.9 m/s. (**c**) *V*_c_ = 26.2 m/s. (**d**) *V*_c_ = 30.4 m/s.

**Figure 14 micromachines-13-01421-f014:**
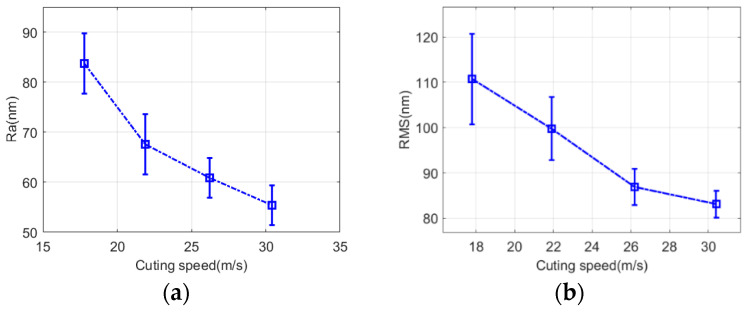
Effect of grinding marks on surface roughness under different grinding wheel linear speeds. (**a**) *Ra*. (**b**) *RMS*. (**c**) *PV*.

**Figure 15 micromachines-13-01421-f015:**
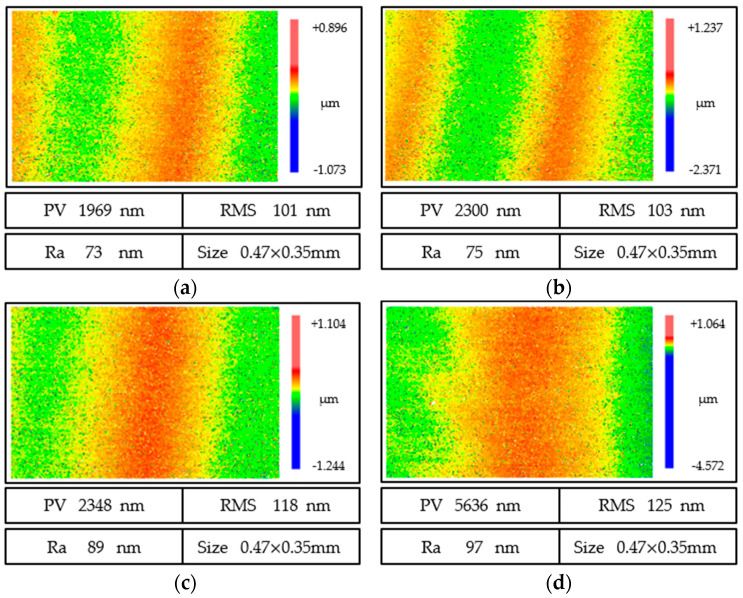
Surface quality evolution of ground fused quartz at different feed rates: (**a**) *V* = 1.5 mm/min; (**b**) *V* = 2 mm/min; (**c**) *V* = 2.5 mm/min; (**d**) *V* = 3 mm/min.

**Figure 16 micromachines-13-01421-f016:**
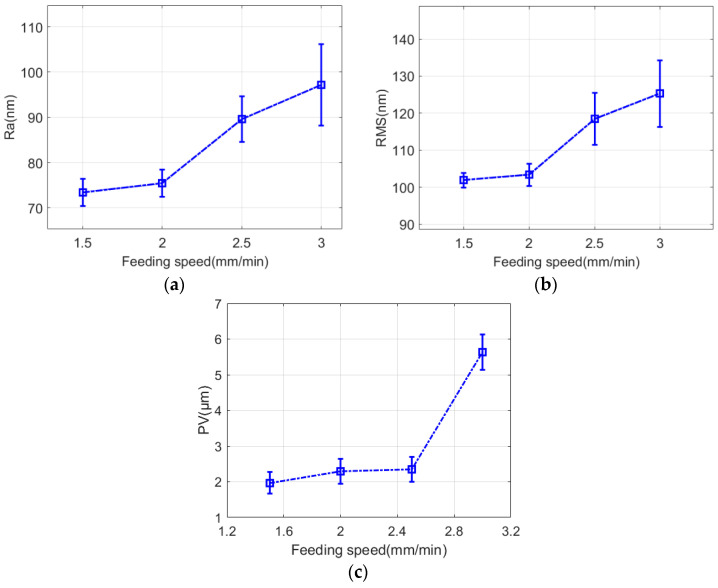
Effect of grinding marks on surface quality at different feed speeds. (**a**) *Ra*. (**b**) *RMS*. (**c**) *PV*.

**Figure 17 micromachines-13-01421-f017:**
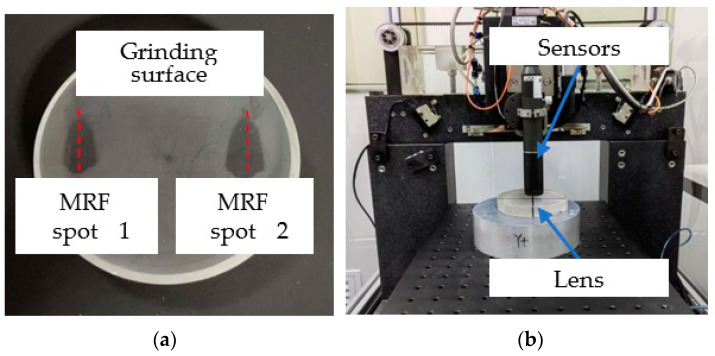
Measurement of subsurface cracks after grinding using a magnetorheological spot method. (**a**) MRF spots on the surface of samples after HF Shallow etching. (**b**) Measuring spot centerline using a three-dimensional profiler.

**Figure 18 micromachines-13-01421-f018:**
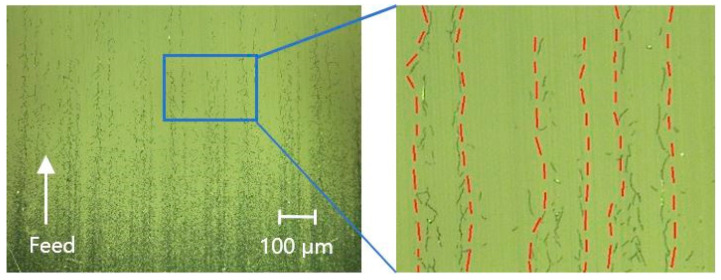
SSD extending along grinding marks.

**Figure 19 micromachines-13-01421-f019:**
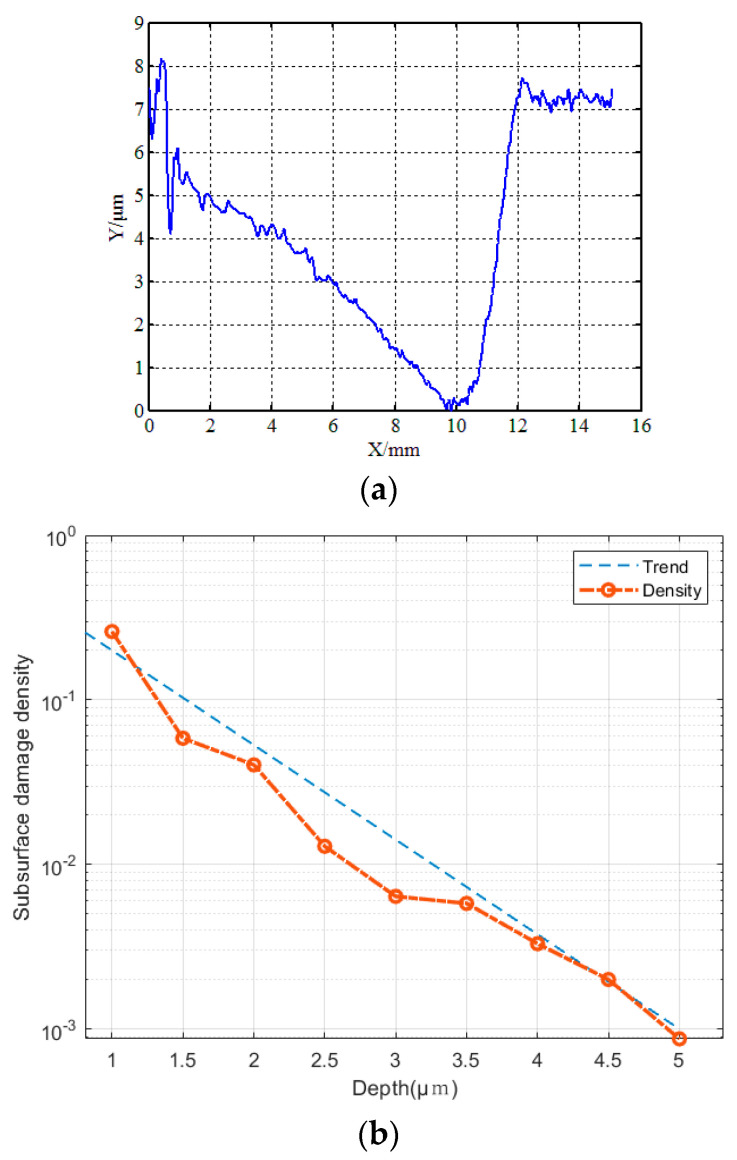
Magnetorheological spot profile and subsurface crack density. (**a**) Depth of profile. (**b**) Crack density distribution.

**Figure 20 micromachines-13-01421-f020:**
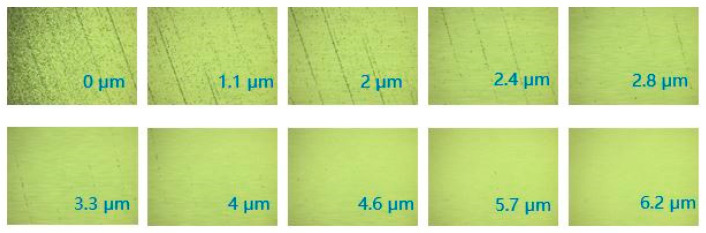
Microscopic images of subsurface cracks at different subsurface depths.

**Figure 21 micromachines-13-01421-f021:**
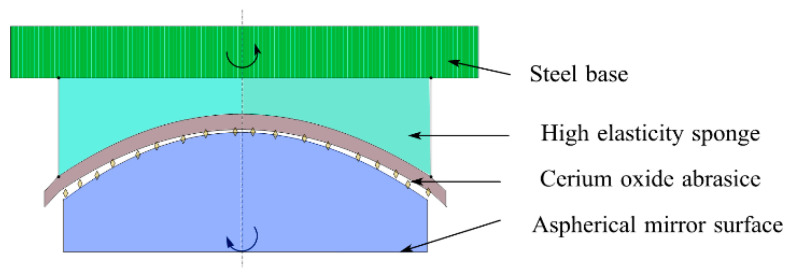
Polishing diagram.

**Figure 22 micromachines-13-01421-f022:**
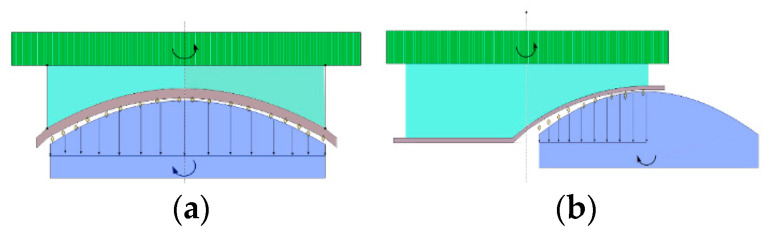
Pressure acting on the workpiece. (**a**) Pressure distribution at the center. (**b**) Pressure distribution at the edge.

**Figure 23 micromachines-13-01421-f023:**
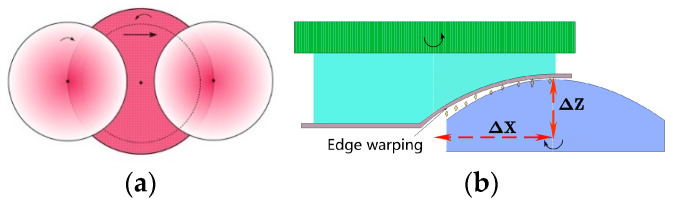
Change in polishing tracking: (**a**) polishing path; (**b**) edge warping.

**Figure 24 micromachines-13-01421-f024:**
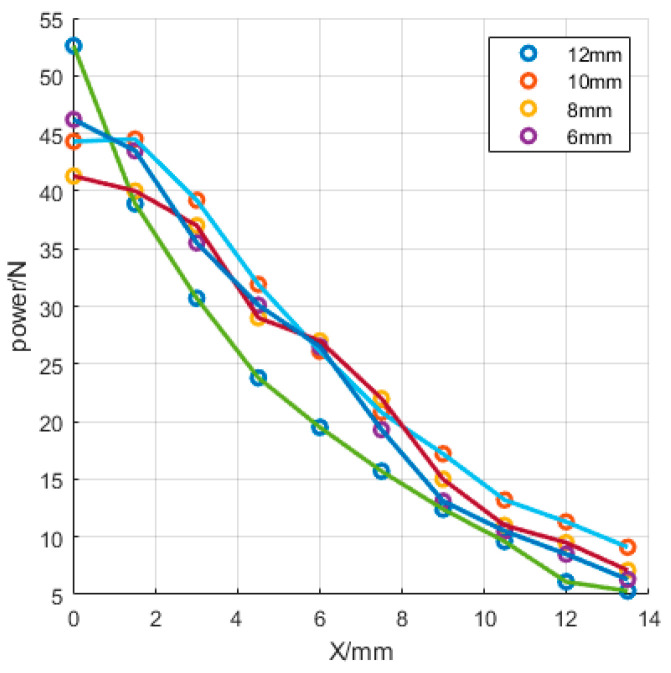
Changes in pressure under different trajectories.

**Figure 25 micromachines-13-01421-f025:**
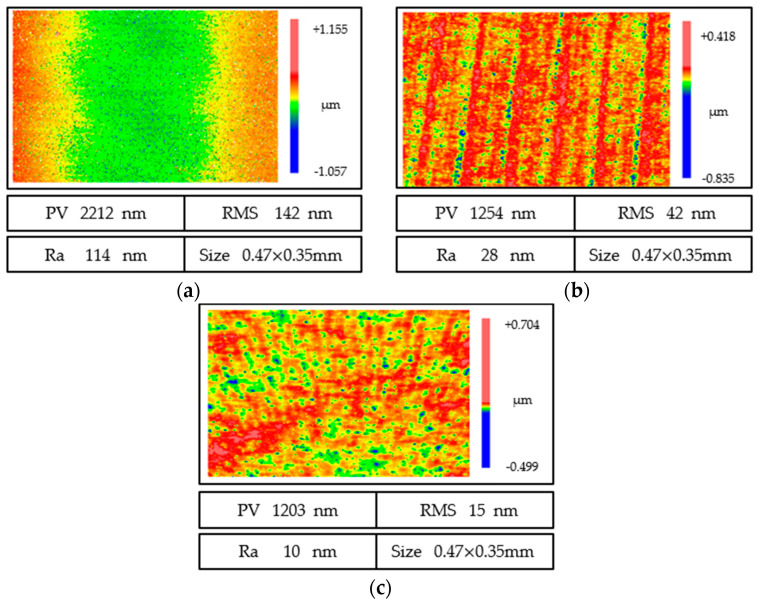
Evolution of surface quality. (**a**) Surface quality after grinding. (**b**) Rough surface quality. (**c**) Surface quality after finishing.

**Figure 26 micromachines-13-01421-f026:**
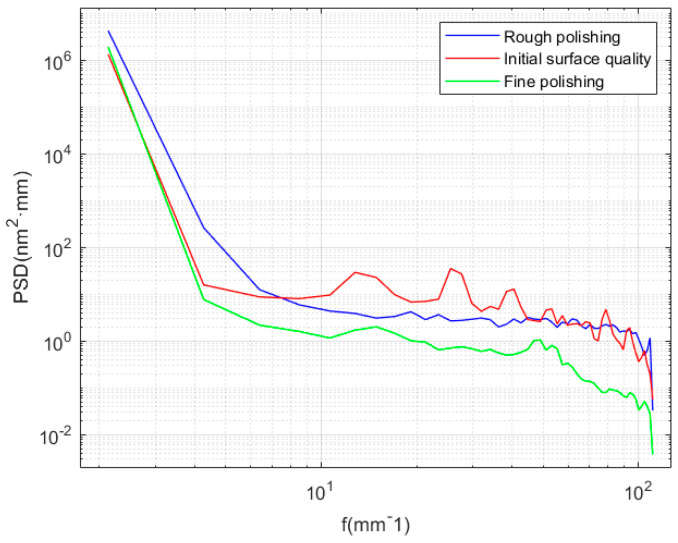
PSD curve under different polishing parameters.

**Table 1 micromachines-13-01421-t001:** Grinding conditions for single-factor tests.

Experiment Number	Grain Size (μm)	Size Range (μm)	Wheel Speed (rpm)	Workpiece Speed (rpm)	Grinding Depth (μm)	Feed Speed (mm/min)
01	D27	25–30	2635	30	5	3
02				40		
03				50		
04				60		
05				70		
06			2721	50		3
07			2871			3
08			2635	50		2
09						4

**Table 2 micromachines-13-01421-t002:** Experimental parameters for single-factor grinding.

Test Number	Grinding Parameters			
	Cutting Depth *A*_e_ (μm)	Grinding Wheel Speed *V*_c _(m/s)	Feed Speed *V* (mm/min)	Grinding Wheel Type
1	1	27.2	2.5	1000# Electroplated grinding wheel
2	3			
3	5			
4	10			
5	1	17.8		2000# Resin grinding wheel
6		21.9		
7		26.2		
8		30.4		
9		27.2	1.5	
10			2	
11			2.5	
12			3	

**Table 3 micromachines-13-01421-t003:** Polishing parameters.

Parameters	Rough Polishing	Fine Polishing
Speed of polishing tool (rpm)	300	300
Workpiece speed (rpm)	30	30
Concentration of polishing solution	50% Cerium oxide	33% Cerium oxide
Polishing path △X (mm)	8	8 mm
Pressing depth (mm)	7	5 mm
Feed speed (mm/min)	60	60
Number of cycles	20	10

## Data Availability

The data presented in this study are available on request from the corresponding author. The data are not publicly available due to the data also forms part of an ongoing study.

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
