# Peer review of "A Method of Restraining the Adverse Effects of Grinding Marks on Small Aperture Aspheric Mirrors"

_micromachines, 2022, doi:10.3390/mi13091421_

Round 1
Reviewer 1 Report
In this manuscript, authors investigated the influence of grinding parameters of in the grinding marks of small aperture aspherical mirrors. The work is novelty and timely. The data is abundant. However, the manuscript was not well-prepared and the presentation of figures need to be improved. I suggest to consider for the publication in Micromachines after major revision.
1. Most of figures need to be improved. Don't use screenshots (Fig. 3, 11, 13, 15, 25). The presentation of data is too arbitrary. Some figures can be merged. The captions of figures are not detailed.
The size and format of sub-figures should be modified.
2. Which kind of subsurface damage affect the optical properties? There is no structural characterization in this study.
3. There are too many typos and errors. Please check the manuscript carefully.
The title is confusing. If I understand correctly, you want to show that the influence of grinding marks on the intermediate frequency error of small aperture aspherical mirrors, and suppress grinding marks by the elastic adaptive polishing method.
4. Fig. 12b should be the RMS as a function of depth of cut, not cutting speed.
5. The improvement of surface quality is realized by changing the abrasive particle from diamond to cerium oxide? However, cerium oxide has already been widely used in polishing of glasses. What's the material removal mechanism during the polishing process by cerium oxide?
6. The influence of surface quality in the optical properties is unclear. The comparison of the optical performance of mirrors after previous and proposed grinding method should be added.
Author Response
Response to Reviewer’s Comments
Reviewers' comments: In this manuscript, authors investigated the influence of grinding parameters of in the grinding marks of small aperture aspherical mirrors. The work is novelty and timely. The data is abundant. However, the manuscript was not well-prepared and the presentation of figures need to be improved. I suggest to consider for the publication in Micromachines after major revision.
Point 1: Most of figures need to be improved. Don't use screenshots (Fig. 3, 11, 13, 15, 25). The presentation of data is too arbitrary. Some figures can be merged. The captions of figures are not detailed.
The size and format of sub-figures should be modified.
Response: Yes, the screenshot data is not clear, the full text has been modified. The description text of some numbers is not detailed, it has been modified and introduced in detail. The following figure is an example:
Figure 11. Evolution of the surface quality of grinding marks under different cutting depths for an electroplated grinding wheel.
Point 2: Which kind of subsurface damage affect the optical properties? There is no structural characterization in this study.
Response: Your comments are very valuable to this article, and there is indeed a lack of the impact of subsurface damage on subsurface damage in this paper. I have made the following supplementary remarks in the text:
In ultra-precision grinding, the maximum depth of subsurface defect of fused quartz material is the median crack caused by brittle fracture. In the process of abrasive scratching removal, the lateral crack caused by the median crack extending to both sides extends to the surface of the specimen, which will lead to the brittleness removal of the material and form the surface roughness on the surface of the specimen. The crack in the grinding process does not necessarily extend to the surface, and the median crack is not visible on the surface. On the other hand, the median crack constitutes subsurface damage. The existence of these damages not only affects the imaging quality, coating quality and service life of quartz glass, but also directly determines the removal amount and processing efficiency of the next polishing process. Rapid and accurate detection of subsurface damage introduced in the grinding process can guide the optimization of processing technology and improve the processing quality and efficiency of quartz glass.
Point 3: There are too many typos and errors. Please check the manuscript carefully. The title is confusing. If I understand correctly, you want to show that the influence of grinding marks on the intermediate frequency error of small aperture aspherical mirrors, and suppress grinding marks by the elastic adaptive polishing method.
Response: There is a problem of inaccurate words in some professional vocabulary, and the full text has been revised. What you said is exactly the work that this paper wants to show, and there is a problem of improper expression in the title. Changes have been made.
The new topic is: A method of restraining the adverse effects of grinding marks on small aperture aspheric mirrors.
Abbreviations are usually defined when they are first used in abstracts and bodies. The author has checked whether all abbreviations in the paper are defined at first use, such as magnetorheological finishing(MRF) in the abstract and the definition of hydrofluoric acid in the picture in figure 17 (a).
There is a problem with some of the statements, which has been corrected. Such as “After oblique axis ultra-precision grinding, the surface roughness 6.8 nm roughness(Ra) and the shape accuracy reached 383 nm peak and valley (PV).”
line276.“Low frequency morphology” should be changed to “low frequency error”.
Line148. The sentence expression is inappropriate. “The radius of the small diameter aspheric surface was L. F was the feed speed in X-direction.”
Point 4: Fig. 12b should be the RMS as a function of depth of cut, not cutting speed.
Response: This diagram has been modified.
Figure 12(b). Effects of grinding marks on surface quality for different cutting depths.
Point 5: The improvement of surface quality is realized by changing the abrasive particle from diamond to cerium oxide? However, cerium oxide has already been widely used in polishing of glasses. What's the material removal mechanism during the polishing process by cerium oxide?
Response: The elastic polishing wheel combined with cerium oxide liquid can restrain the grinding marks, and there are three removal mechanisms in the polishing process. One is pure mechanical removal, which follows the Preston equation, and the plastic removal occurs on the surface of the material after the fine particles are under pressure. The second is chemical removal. Cerium oxide liquid hydrolyzes fused quartz and produces silicic acid gel film. The third is the removal of flow. Except for friction heat, the surface plastic deformation and flow, heat softening to melting and flow, convex filling, surface molecules redistributed to form a flat surface. In the comprehensive process, the three functions exist at the same time.
Point 6: The influence of surface quality in the optical properties is unclear. The comparison of the optical performance of mirrors after previous and proposed grinding method should be added.
Response: Yes, this paper lacks an explanation of the effect of surface quality on optical properties and a comparative introduction before and after polishing, which has been supplemented in the text.
Ultra-precision grinding has played an important role in the processing of optical elements of hard and brittle materials. The high-precision surface shape can be obtained by grinding, which can greatly reduce the machining allowance for subsequent polishing. So as to improve the overall processing efficiency of optical parts. However, grinding marks will be produced after ultra-precision grinding, and the surface quality PV ,RMS and Ra can only reached 2212,142 and 114nm. The use of fine-grained grinding wheel will affect the surface shape accuracy because of the rapid wear of the grinding wheel. Therefore, it is necessary to use elastic adaptive polishing to improve surface quality. After path optimization, liquid optimization and other measures, the surface quality PV,RMS and Ra reached 1203, 16 and 10nm, which greatly improved the surface quality. The imaging quality and coating quality were optimized.
Dear reviewer:
Thank you for your modification suggestions. I revised the writing of the manuscript based on your suggestion again. I also made other modifications:
- Figure 14 (a) shows that it is incomplete. Modifications have been made.
Figure 14. Effect of grinding marks on surface roughness under different grinding wheel linear speeds.
- The format of the drawing notes in this paper has been unified, otherwise it will be too messy.
- Some formulas are not in the middle, such as formula 2, 4, 4, 5.
, (2)
- Figure 6 shows that it is incomplete. Modifications have been made.
(a)r30 (b)r40
(c)r50 (d)r60
Figure 6. Tracking of abrasive particles in the grinding center under different workpiece matching speeds.
- “Intermediate frequency error” is changed to “mid-spatial frequency error”.
It has been modified to the correct format
The specific modifications are in the manuscript.
Best regards!

Reviewer 2 Report
The article is correct in terms of content and concerns current scientific problems related to grinding and polishing of aspherical mirrors. The process of ultra-precise grinding and polishing of molten quartz material was correctly analyzed and the effect of grinding marks was analyzed. The conclusions are correct. Literature selected properly. These are the positive aspects of the article. Potentials for improvement:
-) In the abstract, I propose to add a clearly defined purpose of the article.
-) Line 120. The last task before the formula (1) should end with a colon (:).
-) Is the scale shown in Fig. 5a, is the same for parts a and b?
-) I suggest not to end subchapters (e.g. 2.3, 4.3) with figure. I propose to end the subsections with the text of the commentary.
Positive: This work was financially supported by the National Key Research and Development 637 Program of China (No. 2019YF0708903), Fundation of State Key Laboratory of Digital Manufacturing equipment and techology (Grant No. DMETKF2022006) National Natural Science Foundation 639 of China (No. 51835013 , National Natural Science Foundation of China (No. 51835013), Open Project 640 of the State Key Laboratory of High Performance Complex Manufacturing (No.Kfkt2021-07) , 641 National University of Defense Technology Research Program (ZK22-12).
I don't feel qualified to judge about the English language and style. I recommend publishing the article.
Author Response
Response to Reviewer’s Comments
Reviewers' comments: The article is correct in terms of content and concerns current scientific problems related to grinding and polishing of aspherical mirrors. The process of ultra-precise grinding and polishing of molten quartz material was correctly analyzed and the effect of grinding marks was analyzed. The conclusions are correct. Literature selected properly. These are the positive aspects of the article. Potentials for improvement:
Point 1: In the abstract, I propose to add a clearly defined purpose of the article.
Response: Okay, It has been added in the abstract.
In this paper, the source of grinding marks and its influence on the mid-spatial frequency error of small aperture aspheric mirror were analyzed, and the grinding marks was suppressed by elastic adaptive polishing.
Point 2: Line 120. The last task before the formula (1) should end with a colon (:).
Response: Okay, It has been modified..
Point 3: Is the scale shown in Fig. 5a, is the same for parts a and b?
Response: a is not the same as b. There is a lack of scale in b. The scale has been added to figure 5 (b) in the body picture.
Figure 5. Periodic ring breaking.
Point 4: I suggest not to end subchapters (e.g. 2.3, 4.3) with figure. I propose to end the subsections with the text of the commentary.
Response: Your comments are very valuable, the ending with the picture is very abrupt, All the chapters of the full text end with a commentary.
Positive: This work was financially supported by the National Key Research and Development 637 Program of China (No. 2019YF0708903), Fundation of State Key Laboratory of Digital Manufacturing equipment and techology (Grant No. DMETKF2022006) National Natural Science Foundation 639 of China (No. 51835013 , National Natural Science Foundation of China (No. 51835013), Open Project 640 of the State Key Laboratory of High Performance Complex Manufacturing (No.Kfkt2021-07) , 641 National University of Defense Technology Research Program (ZK22-12).
I don't feel qualified to judge about the English language and style. I recommend publishing the article.
Response: Thank you for your recognition of the work of this article.
Dear reviewer:
Thank you for your modification suggestions. I revised the writing of the manuscript based on your suggestion again. I also made other modifications:
- Figure 14 (a) shows that it is incomplete. Modifications have been made.
Figure 14. Effect of grinding marks on surface roughness under different grinding wheel linear speeds.
- The format of the drawing notes in this paper has been unified, otherwise it will be too messy.
- Some formulas are not in the middle, such as formula 2, 4, 4, 5.
, (2)
- Figure 6 shows that it is incomplete. Modifications have been made.
(a)r30 (b)r40
(c)r50 (d)r60
Figure 6. Tracking of abrasive particles in the grinding center under different workpiece matching speeds.
- “Intermediate frequency error” is changed to “mid-spatial frequency error”.
It has been modified to the correct format
The specific modifications are in the manuscript.
Best regards!

Round 2
Reviewer 1 Report
The comments have been addressed. But the figures still need yo be modified.
The images in each row of Figure 2 are not aligned;
The size of images in Fig. 6 is different;
Fig. 9, ab have grid lines in the background, cd not;
....................